# From Design to Study of Liposome-Driven Drug Release Part 1: Impact of Temperature and pH on Environment

**DOI:** 10.3390/ijms241411686

**Published:** 2023-07-20

**Authors:** Violetta Kozik, Danuta Pentak, Marlena Paździor, Andrzej Zięba, Andrzej Bąk

**Affiliations:** 1Institute of Chemistry, University of Silesia, Szkolna 9, 40-006 Katowice, Polandandrzej.bak@us.edu.pl (A.B.); 2Faculty of Chemistry, University of Opole, Oleska 48, 45-052 Opole, Poland; 3Department of Organic Chemistry, Faculty of Pharmaceutical Sciences in Sosnowiec, Medical University of Silesia in Katowice, Jagiellońska 4, 41-200 Sosnowiec, Poland; zieba@sum.edu.pl

**Keywords:** nanocarriers, liposomal delivery systems, encapsulation efficiency, drug release profile

## Abstract

The marketed drug Doxorubicin (DOX) and the promising anti-cancer agent 9-(*N*-piperazinyl)-5-methyl-12(*H*)-quino[3,4-*b*][1,4]benzothiazinium chloride (9-PBThACl) were used to prepare and compare a range of liposomal delivery systems based on dipalmitoylphosphatidylcholine (DPPC). Liposome-assisted drug release was examined using the spectrophotometric method. In order to provide in vitro release characteristics of liposomal conjugates (L_DPPC/drug_ vs. L_DPPC/drug/drug_) as well as to evaluate the impact of temperature and pH buffering on the conformation/polarity of the phospholipid bilayer, the encapsulation efficiency of the liposomes entrapping 9-PBThACl and DOX was calculated. In fact, some competition between the investigated molecules was noticed during the entrapment process because relatively high values of the encapsulation efficiency were observed only for the liposomal complexes containing one trapped drug molecule. An averaged absorbance value enabled us to indicate the pH value of the environment (pH ≈ 6.8), at which the physicochemical property profiles of the liposomal complexes were noticeably changed. Moreover, the operational factors limiting the drug release kinetics from the produced liposomes were mathematically modeled. First-order and Bhaskas models ensured satisfactory compliance with the experimental data for the liposomal complexes buffered at pH values of 5.50, 6.00, and 7.40, respectively.

## 1. Introduction

The development of innovative delivery systems based on biomaterials and nanotechnology might enhance not only a drug’s stability and bioavailability but also diminish the toxicity of the potential pharmaceutical agent. Hence, an intensively growing interest has been observed recently in targeted therapy using nanoparticles as carriers of active substances that facilitate drug transport from the place of administration to the site of action [1]. One of the most commonly used nanoscale carriers for the targeted delivery of drugs are liposomes—artificially structured spherical vesicles composed of a phospholipid bilayer enclosing an aqueous solution [2,3,4,5]. It seems that the potential penetration of liposome-encapsulated anti-cancer substances into the surrounding tissues results from the fact that the endothelium of tumor blood vessels and healthy cells varies notably [6]. The injection of the drug-containing liposome allows for a decrease in the dose of the active substance and a reduction in the frequency of administration, limiting the overall drug toxicity while preserving the desired ADMET-like properties. In other words, the drug’s biodistribution (pharmacokinetics and pharmacodynamics) can be improved noticeably since the liposomal form of the drug stays longer in the bloodstream [7]. Overall, recent years have witnessed the application of liposomes as nanocarriers for anti-viral, anti-fungal, anti-diabetic, and anti-cancer agents, enhancing their pharmaceutical effectiveness [8,9,10,11]. It should be highlighted that the entrapment of medicines can also be conducted using other nanocarriers. For instance, exosomes and niosomes are the nanostructured media that are used in the transportation of chemotherapeutics [12] and sometimes compose hybrid structures with liposomes to facilitate the distribution of hydrophobic substances [13]. Moreover, derivatives of graphene oxide were also proposed to embed nanoparticles in potential drug delivery systems [14].

Regardless of the administration route, pharmaceutical availability is a significant factor defining the phase of the active component release from the drug molecule. In technological practice, the release study has become indispensable to assess a drug’s quality at the early stages of formulation preparation [15]. In order to prove the reproducibility of the manufacturing operations and the steady composition of each bunch of the final product, the drug release evaluation has to be investigated meticulously, especially after a drug’s registration. Moreover, the operational parameters of a drug’s production can be controlled systematically. A comprehensive analysis of the release profile might provide valuable hints in cases where a drug’s properties are unexpectedly changed, whereas other methods might prove inefficient.

In practice, a release study can be conducted using the one-point and multi-spot tests, respectively. In the one-point approach, the quantity of the released substance (Q) is compared to the volume of the acceptor liquid as a function of time; the resulting ratio is referred to as the previously specified acceptance criteria. In the multi-point test, the variations of the released substance are recorded at a few or even numerous time spots. Armed with such data, the drug release profiles might be established in the form of curves presenting the changes of the released substance as a function of time. Not surprisingly, in-depth drug release profiling should be routinely performed at the early stages of drug R&D activities since the multi-point examination is more precise compared to the single-spot test.

In the presented study, the marketed drug Doxorubicin (DOX) and the promising anti-cancer agent 9-(*N*-piperazinyl)-5-methyl-12(*H*)-quino[3,4-*b*][1,4]benzothiazinium chloride (9-PBThACl) were utilized to prepare a range of liposomal complexes (L_DPPC/drug_ vs. L_DPPC/drug/drug_). In fact, the anti-cancer potential of neuroleptic phenothiazines containing alkylaminoalkyl substituents at the thiazine nitrogen atom has been reported previously [16]. It seems that the anti-proliferative activity of 9-PBThACl reagents stems from an interaction between such structural motifs and enzymes involved in inducing apoptosis, as well as from DNA intercalating properties that induce its fragmentation in cancer cells. Similarly, to the action exerted by the anthracycline antibiotics, the planar structural fragments of the analyzed quinobenzothiazine salts can ease intercalation into the DNA helix. The stability of the DNA-drug complex can be enhanced by the existence of an intercalating factor (a substitute capable of forming hydrogen bonds with purine and pyrimidine bases) that raises the stability of the DNA-drug conjugate, resulting in inhibited cell proliferation. In this context, the experimental and theoretical findings of the drug release modeling using six selected mathematical models implemented to describe the release profiles of a new, biologically tested 9-PBThACl molecule and doxorubicin appear valid. Overall, the obtained data proved the immense contribution of the pH buffer in the environment to the physicochemical profiles of the prepared liposomal conjugates. In fact, the pH value of the environment was specified, at which the physicochemical property profiles of the liposomal complexes were noticeably changed. It was also demonstrated that the rise in temperature resulting in the increment of the phospholipid bilayer fluidity can damage the membrane hydrogen bonds accordingly. Moreover, some competition between the investigated molecules was noticed during the entrapment process because relatively high values of the encapsulation efficiency were recorded only for the liposomal complexes containing one trapped drug molecule. Additionally, the operational factors that limit the rate of drug release from the produced liposomes were modeled mathematically.

## 2. Results and Discussion

Nanoparticle tracking analysis (NTA) is the method commonly used for the physical characterization and quantification of extracellular vesicles in biological samples in suspension in the range from 10 to 1000 nm based on the analysis of Brownian motion [17,18,19]. Hence, the critical evaluation of L_DPPC/DOX_, L_DPPC/9-PBThACl_, and L_DPPC/9-PBThACl/DOX_ liposomes was performed using the NTA methodology in order to specify the size distribution of the investigated liposomal complexes as potential drug carriers. A limited number of huge liposomal aggregates (size > 300 nm) does not affect the average size distribution of the analyzed complexes, as shown in Figure 1.

### 2.1. Encapsulation Degree Analysis

The examination of the liposomal encapsulation degree of 9-PBThACl and Doxorubicin hydrochloride was conducted using the spectrophotometric methodology for systems with a surrounding pH changing from 5.50 to 7.40. The absorbance measurements were performed at 488 nm for 9-PBThACl and at 233 nm for DOX, respectively. The standard curves of the absorbance against the concentration were plotted for the examined drugs. Consequently, the concentration of the nano-encapsulated drug was specified in the supernatant. The encapsulation efficiency (*EE*%) of the liposomes entrapping 9-PBThACl and DOX was calculated as the mass ratio between the amount of the drug incorporated in the liposomes and the ratio used in the liposome preparation according to Equation (1). Table 1 and Table 2 report the encapsulation efficiencies of 9-PBThACl and DOX in the liposomes, respectively.

The drug loading (DL%) was calculated for different liposomal systems according to Equation (2) for the pH values of 5.50, 6.00, 6.50, and 7.40, as shown in Table 3 and Table 4.

The presented findings confirmed the impact of buffer pH on the encapsulation degree of the investigated molecules.

### 2.2. Stability of Liposomal Preparations

The evaluation of physical and chemical stability is the principal research objective in the investigation of novel potential carriers for therapeutic compounds. The provided data form the basis for defining the conditions of technological conduct, the specification of auxiliary agents in the development of drug formulations, and the selection of their storage conditions. The durability of medicinal substances themselves and their transport structures are directly dependent on physicochemical factors [20]. The 5 week time period was chosen to analyze the stability of the prepared liposomal aggregates, during which the samples were stored at a temperature of 4 °C. The weekly recorded drug release rates (*R*%) calculated according to Equations (3) and (4) for varied liposomal systems are illustrated in Figure 2(A1–D2).

The greatest variations in the degree of the analyzed drug release were observed at a pH level of 6.5 for the liposomal complexes L_DPPC/9-PBThACl/DOX_ vs. L_DPPC/9-PBThACl_ and L_DPPC/9-PBThACl/DOX_ vs. L_DPPC/DOX_, reaching an average of ~28–30% for 9-PBThACl and ~5% for DOX, respectively. For the same pH value in the environment, the percentage of DOX and 9-PBThACl encapsulation in the liposomal L_DPPC/9-PBThACl/DOX_ conjugate is also smaller (see Table 2), gaining 22.16% for 9-PBThACl and 40.37% for DOX. According to our previous findings, the release percentage of the liposome-encapsulated drugs depends on both the surrounding pH and the entrapment site in the nanocarriers [5]. In general, the hydrophilic compounds are located in the central, buffered zone of the liposomes, while the hydrophobic ones are incorporated in the phospholipid bilayer [21]. Consequently, the drug arrangement itself within the liposome structure determines the degree of its release. As a matter of fact, the molecules entrapped in the hydrophilic zone of the liposomes need to penetrate the phospholipid bilayer; therefore, the liposomal membrane has to be in the liquid (crystalline) phase, which is disordered and leaky [22]. The fluidity of the bilayer is related to many factors [23], primarily: (**i**) the character and value of the electrostatic charges of phospholipid ‘heads’ composing the liposomal structure; (**ii**) the length of fatty acid chains that are present in phospholipids; (**iii**) the number of unsaturated bonds in phospholipids; and (**iv**) the concentration of cholesterol and protein molecules. Due to the lack of covalent bonds, the liposomal bilayer is not a stiff structure. In fact, twofold interactions between the membrane building blocks are observed that outline the surface (polar) area as well as the inner (hydrophobic) zone, which is characterized mainly by dispersion interactions. 

The accidental fluctuation of the electron density can occur in molecules without a permanent dipole moment, resulting in the appearance of the instantaneous dipole moment despite the lack of noticeable bond polarizations. The emerging force field surrounding a molecule can induce the polarization of the neighboring molecule with the generation of another dipole moment. The above phenomenon may happen within the entire bilayer simultaneously between the interacting molecules. It is believed that dispersion forces decrease the energy of the system thanks to the correlated fluctuations of charge distributions in the proximal molecules. The power of dispersion interactions reaches the maximal values for molecules that are set parallel to each other; therefore, Van der Waals forces strive towards the parallel arrangement of hydrocarbon chains—the closer the better, since the dispersion energy is inversely proportional to the distance r between the interacting molecules $\left(r^{-6}\right)$. In addition, the density of phospholipid packing depends on the proportion between the ‘head’ size and the remaining hydrophobic part of the molecule [24]. The hydrophilic areas of the phospholipid bilayer constitute the polar lipids, interacting as ion-ion, ion-dipole, dipole-dipole, or hydrogen bonds (HB). Moreover, the interface boundary with the absorbed water molecules (called the structural water) is also a significant component of the phospholipid bilayer [25], particularly for glycolipids that form the energetic barrier for transporting the hydrophobic molecules. Due to the strong ionization of the bilayer, other ions and ionized molecules present in solution are adsorbed on the membrane surface, particularly those not transported through the lipid boundary. The strong adsorption properties of the multivalent ions play a valid role in organizing polar (particularly ionized) parts of phospholipids situated on the membrane interface (called phospholipid ‘head’ cross-linking). Doxorubicin, as well as 9-PBThACl, are hydrophilic molecules. It seems that DOX is probably bonded with the polar bilayer via relatively weak interactions. The resulting disorganization of the membrane polar area facilitates the release of 9-PBThACl molecules that are localized in the central hydrophilic part of the liposome (see Figure 3). Each modification of the bilayer structure involving the introduction of another membrane component (even localized on the surface) directly effects the process of drug release [26]. Undoubtedly, the surface area of the bilayer is increased, resulting in changes in the membrane’s physicochemical properties.

### 2.3. Influence of Surrounding pH on Drug Release–Spectrophotometric Analysis

The preparatory procedures of liposomal systems were conducted so as to maintain the constant ratio of the analyzed compounds to phospholipid, amounting to 30:1:1 for L_DPPC/9-PBThACl/DOX_; and 30:1 for L_DPPC/9-PBThACl_; 30:1 for L_DPPC/DOX_, respectively. The detailed analyses of the absorption spectra obtained for L_DPPC/9-PBThACl/DOX_, L_DPPC/9-PBThACl_, and L_DPPC/DOX_ allowed us to evaluate the changes appearing in the liposomal complex with two molecules in tandem (L_DPPC/9-PBThACl/DOX_), as shown in Figure 4 and Figure 5. The following assumptions were made:(a)A = 0, if subtracting the L_DPPC/9-PBThACl/DOX_ spectrum from the L_DPPC/9-PBThACl_ or subtracting the L_DPPC/9-PBThACl/DOX_ spectrum from the L_DPPC/DOX_ yields zero absorbance, that means there are no interactions between the analyzed compounds;(b)∆A ≠ 0, if subtracting the L_DPPC/9-PBThACl/DOX_ spectrum from L_DPPC/9-PBThACl_ or subtracting the L_DPPC/9-PBThACl/DOX_ spectrum from L_DPPC/DOX_ produces non-zero absorbance that suggests interactions within the investigated complex;(c)∆A > 0 and ∆A < 0 are particularly relevant because:∆A > 0, when the absorbance of molecule X in tandem is higher than the absorbance of X alone, is regarded as a reference value. Then, the growth of absorbance is the result of the increased release degree of molecule X influenced by the presence of compound Y in the binary system;∆A < 0, when the absorbance of molecule X in the binary complex is lower compared to the corresponding A value observed for the singular complex treated as a reference. In this case, the absorbance decrease results from the reduced release degree of X affected by the appearance of molecule Y in tandem.


Agreed assumptions explain the competitive effect, while 9-PBThACl is incorporated and then released from the phospholipid nanocarriers.

According to our previous findings, the fluidity of the phospholipid bilayer increases with the rise of the surrounding pH [27]. The hydrogen bonds are probably formed between the oxygen atom of the phosphate group in phospholipids and the hydrogen atom of the hydroxyl group present in the proximal DPPC molecule. If the pH of the environment goes beyond the value of 7.40, then the hydrogen bonds existing in the acidic environment are broken down. As a consequence, an increment in the bilayer fluidity is observed. Stiffened so far with the hydrogen bonds, the polar phospholipid ‘heads’ begin to move, imparting the characteristic fluidity of the membrane. In the investigated range of pH values (from 5.50 to 7.40), the noticeable adjustment of the membrane fluidity directly affects the process of drug release for the analyzed set of compounds. Mathematical fitting of a function to an average score of the absorbance recorded in the subsequent weeks of the liposome incubation, which differs from the pH value (see Figure 6), is comparable with the curve illustrating the impact of pH on the value of the N parameter obtained via the analysis of the phospholipid membrane using EPR spectroscopy [5,28]. Compared to pH = 7.40, the fluidity of the phospholipid bilayer is lower in the range of pH 5.50–6.50. It appears that the perpendicular movement of 9-PBThACl towards the membrane surface along the aliphatic phospholipid chains is feasible thanks to the evolving packing of phospholipids as well as the growing relaxation of the bilayer associated with the increasing pH of the surroundings. A similar mobility of 5-DOXYL spin labels was reported in our previous findings [27]. Moreover, the presence of DOX in the bilayer can directly affect the penetration of 9-PBThACl into the surrounding tissues—depending on the pH of the environment; it may strengthen or weaken the process of 9-PBThACl release accordingly. Possibly, DOX is partly immobilized in the hydrophilic zone of the bilayer using the hydrogen bonds formed with DPPC molecules. As a consequence of such a location, the compact structure of the bilayer is noticeably disturbed, which facilitates the permeation of the membrane via the unbonded 9-PBThACl molecules. The recorded inversions of 9-PBThACl and DOX release characteristics manifest at the potential pH cut-off value (pH ≈ 6.8), as illustrated in Figure 6A and Figure 6B, respectively.

### 2.4. In Vitro Drug Release Examination

Pharmaceutical availability is the quantity of an active component released from a pharmaceutical formulation and dissolved in the surrounding body fluid that is measured under laboratory conditions [29]. Briefly speaking, pharmaceutical availability might be equated with the rate of the drug release process that specifies the parameter characterizing the first stage in the system of drug transformation in the body (LADME). Numerically, the value is calculated as the percentage or ratio of the drug dose that is dissolved in the received fluid as a function of time; however, only the free fraction of the therapeutic compound is absorbed, exerting the corresponding pharmacological effect. Hence, the drug’s active substance must be released from the structure of the drug (its form) in a timely manner. Accordingly, the appropriate drug formulation needs to be constructed a priori. Obviously, the form of the drug and the method of administration primarily govern the rate and degree of active ingredient release. Moreover, the procedure of medicine preparation and the excipients applied are crucial for the release process; therefore, the search for new alternative forms of drug transport seems valid.

According to the FDA guidelines, two basic approaches were proposed for dissolution testing of immediate drug release: the independent and the model-related ones which were successfully implemented in the statistical analysis of the medicine leakage [30]. It should be emphasized that a range of statistical methodologies can be proposed for the quantitative modeling of drug release from controlled delivery systems [31,32,33,34]; however, the first and most widespread one is the concept of similarity and difference factors. Moreover, the mathematical testing and comparison of dissolution profiles can be based on multidimensional (mD) data space analyses [31]. Actually, the FDA guidelines do not recommend the specific modeling procedure, but rather the general objectives are provided to be met as follows: (**i**) the acceptance criterion should be designated so that it is comparable with the data variance calculated for the reference sample and suitable for distance multidimensional specification; (**ii**) the multidimensional distance between the averaged values of the subsequent time points for the reference as well as the test data should be calculated; (**iii**) the 90% confidence level should be determined for the multidimensional distance; and (**iv**) the calculated limit should be compared with the acceptance criterion. If the upper limit of the confidence interval is smaller or equal to the acceptance criterion, the similarity between the two profiles is statistically confirmed. The third group of methods suggested by the FDA for the comparison of in vitro dissolution profiles are model-dependent procedures, where the shape of the profile is specified using a theoretical curve [35]. At least a few such functions are recommended for fitting the experimental data; however, a continuous Weibull probability distribution is commonly applied. 

In practice, six mathematical models were used, including First-order, Bhaskas, Higuchi, Ritger–Peppas, Korsmeyer–Peppas, and an extension of the classical Freundlich model (see Equations (5)–(10)) in order to explain the process of releasing encapsulated compounds from liposomes composed of dipalmitoylphosphatidylcholine as the main component of the membrane [32,36]. The release rate studies were carried out at 37 °C and 41 °C, as shown in Figure 7 and Figure 8, respectively.

As a matter of fact, the presented data confirm the tendency reported previously in the literature, where the direct impact of surrounding pH and temperature on the drug release profiles from liposomes was observed [37]. Overall, a wide range of aspects related to the drug’s in vivo spreading are implemented in the mathematical models, calculated according to Equations (5)–(10). A comprehensive review, including the systematic analysis of different drug release systems in terms of mechanisms that limit the transport of the active component, can be found elsewhere [33,38]. Accordingly, the drug release systems are controlled by diffusion (within the material or through the membrane), degradation of a carrier, material bulking, as well as environmental factors, e.g., pH and temperature variations. Nevertheless, the extensive list of operational parameters that should be taken into account during the search for clarification of the drug release mechanisms is definitely longer [33].

It was revealed that the course of profiles obtained at a temperature of 37 °C and the surrounding pH equal to 5.50, 6.00, and 7.40 were approximated exponentially, as reported in Table 5. Conversely, at pH = 6.5, the drug release process was estimated via linear dependency since 3 out of 5 models (Ritger–Peppas, Korsmeyer–Peppas, and an extension of the classical Freundlich model) were characterized by R^2^_adj_ = 1. An extension of the classical Freundlich model was applied for decreasing functions at a temperature of 41 °C (see Table 5). Undoubtedly, the rise of the temperature to the point of the main DPCC phase transition [39] destabilizes the investigated liposomal systems [40]. It seems that the surrounding pH can directly alter the thermal characteristics of liposome-encapsulated drug complexes.

## 3. Materials and Methods

### 3.1. Materials

1,2-Dipalmitoyl-*sn*-glycero-3-phosphocholine (DPPC, ≥99%), dichloromethane, chloroform, potassium phosphate dibasic (≥98%), sodium phosphate monobasic monohydrate (≥98%), and Doxorubicin hydrochloride (98.0–102.0% (HPLC)) were purchased from Sigma Aldrich, Schnelldorf, Germany. 9-(*N*-piperazinyl)-5-methyl-12(*H*)-quino[3,4-*b*][1,4]benzothiazinium chloride (9-PBThACl) was synthesized at the Department of Organic Chemistry, Medical University of Silesia in Katowice, Poland [16].

### 3.2. The Synthesis of 9-PBThACl

The proposed method of quinobenzothiazine tetracyclic derivative synthesis relies on the cyclization of betaine systems featuring 1-alkyl-4-(arylamino)quinolinium-3-thiolates structure via nucleophilic substitution of the hydrogen or halogen atom in the phenyl ring by a thiolate-derived sulfur atom. Reactions proceed with high yield even at room temperature. Introducing such substituents or functional groups into the benzene ring permits modification of the quinobenzothiazine system, which is difficult to achieve using other methods of synthesis. The presence of oxygen in the reaction mixture has an impact on the course of the reaction between *bis*-chloride and 4-piperazinylaniline (see Figure 1). In the presence of a hydrogen chloride donor and atmospheric oxygen, betaines underwent cyclization to appropriate quinobenzothiazine chlorides. It should be highlighted that the compound can also be synthesized directly from *bis*-chloride using adequate amines in a one-pot reaction in the presence of atmospheric oxygen without separating the intermediate product [16]. The synthesized 9-PBThACl compound shows strong anti-proliferative properties that are comparable with DOX and cisplatin.

### 3.3. Liposome Preparation

The modified reverse-phase evaporation method (mREV) was employed to prepare the liposomes. The mREV procedure relies on the continuous mixing of specific volumes of suitable phospholipid solutions in excess of organic solvents with water according to the methodology described in detail by Papahadjopoulos [41]. The replacement of the ultrasonic dispersion of the aqueous phase in the organic phase with mechanical dispersion that allows complete conversion of the phospholipids to the liposomes is a modification to the above method. Liposomes (L_DPPC_; L_DPPC/9-PBThAC_; and L_DPPC/9-PBThACl/DOX_) were prepared by the mREV method using the DPPC:drug molar ratio 30:1 and the DPPC:drug:drug molar ratio 30:1:1. The lipid dispersion at a final lipid concentration of ca. 2.64 × 10^−2^ M was used. Finally, a total of 0.34 mL of 9-PBThACl and Doxorubicin at a concentration of 5 × 10^−3^ M were added to the prepared mixture. In order to unify the dimensionality of the analyzed structures, the standard method of homogenization was applied using the Avanti mini extruder with a 100 nm filter.

### 3.4. Solutions and Sample Preparation

Phosphate-buffered saline (PBS) pH 5.50, 6.00, 6.50, and 7.40 were used for the UV/Vis measurements. All mixtures used in the reactions were prepared in triplicate to allow for statistical analysis.

### 3.5. The Liposome Size Measurement

Diameters of L_DPPC/DOX_, L_DPPC/9-PBThACl_, and L_DPPC/9-PBThACl/DOX_ liposomes were measured at room temperature with a NanoSight NS300 instrument (NanoSight, Malvern, UK) and subjected to nanoparticle tracking analysis (NTA), yielding size distribution and concentration (particles/mL). The instrument was equipped with a CMOS camera (Hamamatsu Photonics, Hamamatsu, Japan) and a 488 nm laser. All experiments were made in triplicate. The software used for capturing and analyzing the data was NTA 3.2 Dev Build 3.2.16.

### 3.6. Stability Study

The period of 5 weeks was chosen to analyze the stability of the prepared liposomal aggregates, during which the samples were stored at a temperature of 4 °C.

### 3.7. UV/Vis Measurements

The absorption spectra of free 9-PBThACl, Doxorubicin, and the liposomal form of 9-PBThACl and Doxorubicin were recorded with the Lambda Bio 40 spectrometer (Perkin Elmer, Waltham, MA, USA), equipped with a PTP-1 Peltier System (Perkin Elmer, Waltham, MA, USA) automatic temperature controller. The temperature was maintained at 37 ± 0.1 °C and 41 ± 0.1 °C. All the spectra were recorded after the equilibration of the samples with the automatic temperature controller. The spectral analysis was conducted using UV WinLab Perkin Elmer Software (The Lambda Series).

### 3.8. Encapsulation Efficiency and Drug Loading

In order to separate the encapsulated drug from its unencapsulated form, the samples were dialyzed immediately after preparing the liposomal forms of 9-PBThACl and DOX (L_DPPC/9-PBThACl/DOX_). Aliquots (8 mL) of the liposomal form of the analyzed drug dispersion were placed into a Float-A-Lyzer G2 (Spectra/Por) cellulose ester dialysis tubing (Spetra/Por^®^, 8–10 kilodaltons MWCO, Spectrum Laboratories, Inc., Rancho Dominguez, CA, USA), immersed in 50 mL buffer at 4 °C with magnetic stirring at 360 rpm. Then the samples taken from the recipient solution at predetermined times were replaced with the same volumes of fresh buffer and designated spectrophotometrically at 233 nm for the DOX and at 488 nm for the 9-PBThACl ingredient. The standard curve of the drug was used to analyze the concentration of the encapsulated drug by measuring the maximum absorption of the DOX (λ_max_ 233 nm) and of the 9-PBThACl (λ_max_ 488 nm). The encapsulation efficiency (*EE*%) of 9-PBThACl and DOX entrapped within the liposomes was calculated according to Equation (1) [42,43]:(1)$$EE\%=\frac{C_{total}-C_{free}}{C_{total}}\times 100.$$The drug loading (*DL%*) was calculated based on Equation (2) [44,45]:(2)$$DL\%=\frac{C{\;}_{total}-C_{free}}{C_{total\;lipid}}\times 100,$$
where:*C_total_*—total concentration of the drug;*C_free_*—concentration of free drug in the supernatant;*C_total lipid_*—total concentration of lipid.

### 3.9. Drug Release and Mathematical Modeling Study

The examination of 9-PBThACl and DOX release from liposomes (L_DPPC/9-PBThACl/DOX_) was performed at 37 °C and at 41 °C for 5 h. Spectrophotometry was used to determine the degree of drug release from liposomes. During the initial 60 min, the measurements were conducted at 5 min intervals. Afterwards, the measurements were recorded every 15 min. Drug release (*R*%) was calculated according to Equations (3) and (4) [46]:(3)$$R\left(\%\right)=\frac{\left([x{]}_{f\;}-[x{]}_{fo}\right)}{(\left[x{]}_{t}\times EE\%\right)}\times 100,$$
where:

R(%)—encapsulated drug percentage of leakage;

${\left[x\right]}_{f\;}$—concentration of leaked drug;

${\left[x\right]}_{fo}$—initial concentration of unencapsulated drug;

EE%—percentage of drug encapsulation.
(4)$${\left[x\right]}_{t\;}=\frac{{\left[x\right]}_{to}}{\beta }\;,$$
where:

${\left[x\right]}_{to}$—is the concentration of total drug in the original liposomes;

β—dilution.

The following mathematical models were implemented in order to theoretically approach the process and degree of drug release from the liposomes [47,48,49]:(5)$$\mathrm{First}\text{-}\mathrm{order}:\;X=1-e^{-k\left(t-\alpha \right)},$$
(6)$$\mathrm{Bhaskas}:\;X=1-e^{-k{\left(t-\alpha \right)}^{0.65}},$$
(7)$$\mathrm{Higuchi}:\;X=k{\left(t-\alpha \right)}^{0.5}\;,$$
(8)$$\mathrm{Ritger}–\mathrm{Peppas}:\;X=k{\left(t-\alpha \right)}^{n},$$
(9)$$\mathrm{Korsmeyer}–\mathrm{Peppas}:\;X=kt^{n},$$
(10)$$\mathrm{and}\;\mathrm{an}\;\mathrm{extension}\;\mathrm{of}\;\mathrm{the}\;\mathrm{classical}\;\mathrm{Freundlich}\;\mathrm{model}:\;X=a+kt^{n}\;,$$
where:

*X*—release percentage *R*(%) from the fitting of experimental data to mathematical formulas;

*t*—release time;

*k*—kinetic constant;

*α*—modified parameter;

*a*—the fit factor;

*n*—exponent describing the various mechanisms of release.

The value of *n <* 0.45 corresponds to Fick’s diffusion, and the square root of the time value is then proportional to the amount of fraction released from the carrier. A process other than Fick’s diffusion appears for 0.45 *< n <* 0.89, and drug release occurs as a consequence of both the diffusion and controlled mechanisms [50]. Equations 5–10 were evaluated using the residual sum of squares (*SUM*) and *R^2^_adj_* values.

### 3.10. Statistical Analysis

All experiments and measurements were conducted in triplicate, and the data were reported as the mean ± standard deviation (SD). The obtained findings were analyzed using OriginPro 8.5.0 SR1 software (OriginLab Corporation, Northampton, MA, USA).

### 3.11. Weibull Probability Distribution

The Weibull probability distribution used in modeling the release curves was specified according to the following formula:(11)$$m=m_{\infty }\left[1-e^{-{\left(\frac{t}{t_{d}}\right)}^{\beta }}\right],$$
where:

*m*—release percentage of the active substance in time *t*;

*m_∞_*—release asymptote (*m_∞_* = 100%);

*t_d_—*parameter defining the time scale of the process;

*β*—parameter characterizing the shape of the release curve (if *β* = 1, the function shows exponential, *β* < 1 parabolic, and *β* > 1 sigmoidal growth).

It should be emphasized that *β* parameter stands for the model’s adaptability to almost all characteristics of the drug release profiles. Hence, the *β* parameter is recommended by the FDA guidelines to describe and compare the release profiles. Despite the undeniable advantages, the Weibull probability distribution has its own limits—there are no kinetic foundations for the model due to the lack of a parameter directly linked to the release kinetics. In other words, the underlying kinetics of the drug release cannot be characterized thoroughly, but profiling of the drug release is still provided.

## 4. Conclusions

The correctly prepared methodology of the pharmaceutical availability evaluation, containing the comparison of the drug release profiles, should also provide knowledge describing at least three main domains. First of all, the surveys of the drug release should reflect the variations of the substance’s physicochemical properties resulting in the observed changes in the volume and rate of the active component delivery. Secondly, other procedures for product differentiation should be proposed, considering other excipients as well. Finally, in the case of the demonstrated in vitro/in vivo correlation (IVIVC), the drug release profile should be linked with the substance’s bioavailability. The validity of the drug release studies was emphasized by the FDA and EMA agencies, which published a group of guiding principles to follow.

In this context, we present the findings of the drug release modeling using six selected mathematical models implemented to describe the release profiles of a new, biologically tested 9-PBThACl molecule and the marketed drug (Doxorubicin). Firstly, the modified methodology of reverse-phase evaporation (mREV) was employed to design and experimentally prepare a set of liposome-based drug delivery systems. Moreover, the following models were employed to approximate the empirical data: first-order, Bhaskas, Higuchi, Ritger–Peppas, Korsmeyer–Peppas, and an extension of the classical Freundlich model. As a matter of fact, First-order as well as Bhaskas models, which are based on the Weibull probability distribution, ensured satisfactory compliance with the experimental data for the liposomal complexes buffered at pH values of 5.50, 6.00, and 7.40, respectively. It was also revealed that Ritger–Peppas, Korsmeyer–Peppas, and an extension of the classical Freundlich model were characterized by the highest R^2^_adj_ factor for the liposome/drug system at pH = 6.50. On the other hand, the Higuchi model demonstrated the lowest compatibility with the empirical data, with values of R^2^_adj_ ranging from 0.17448 to 0.97732—a complete lack of conformity was also recorded. In general, the received experimental and modeling data confirm the profound impact of the surrounding pH buffer on the physicochemical profile of the produced liposomal complexes, as was reported previously [51].

It was also shown that the existing hydrogen bonds in the membrane might be broken down according to the rise in temperature, resulting in an increment in the phospholipid bilayer fluidity. It seems that the observed influence of the surrounding pH on the liposomal membrane can stem from the protonation of phosphate groups in the acidic environment and the formation of hydrogen bonds with DPPC molecules. An averaged value of absorbance, calculated as the absorbance difference between the L_DPPC/9-PBThACl/DOX_-L_DPPC/9-PBThACl_ and L_DPPC/9-PBThACl/DOX_-L_DPPC/DOX_ systems, enabled us to indicate the pH value of the environment (pH ≈ 6.8) at which the physicochemical property profiles of the liposomal complexes were noticeably changed. During the entrapment in the liposomes, some competition between the investigated molecules was noticed because relatively high values of the encapsulation efficiency (*EE*%) were observed only for the liposomal complexes containing one trapped drug molecule. For instance, the L_DPPC/9-PBThACl/DOX_ complex at pH 7.40 is characterized by roughly halved encapsulation compared to the L_DPPC/9-PBThACl_ and L_DPPC/DOX_ systems. In consequence, the surrounding pH has a significant effect on the resulting liposomal properties.

As follows, the implemented mathematical models offer:Ability to estimate the concentration changes of the released drug molecules as a function of time;Opportunity to track drug transportation;Determination of factors limiting the rate of drug release;Indication of postulated changes in order to optimize the liposomal drug release profile.

It might be concluded that in silico prediction of the liposome-assisted drug delivery characteristics may significantly reduce the cost and time of the drug carrier formulation.

## Data Availability

Not applicable.

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
