# Peer review of "From Design to Study of Liposome-Driven Drug Release Part 1: Impact of Temperature and pH on Environment"

_ijms, 2023, doi:10.3390/ijms241411686_

Round 1
Reviewer 1 Report
The authors deeply investigated the different kinetic release behaviour of doxorubicin and 9-PBThACl from DPPC-based liposome. Some differences were observed and discussed. Of course, this information could be of interest to select an appropriate anticancer delivery system. However, there is an important aspect that the authors need to evaluate. The design and optimization of drug delivery system should be performed also considering the particle size and zeta potential (doi.org/10.1016/j.colsurfb.2021.112217). Both of these properties influence the cellular uptake and also the stability in physiological environment. Therefore, in my opinion, the have to better characterize their system and results discussed in light of both the particles dimension and superficial charges.
In the Materials and Methods section:
1. The authors sinthetised the 9-PBThACl compound. Where is the structural characterization? In particular the authors have to show the NMR spectra and HPLC mass data.
2. The figure of 9-PBThACl (Fig. 8) should not be present in this section since the it is already shown in the Scheme 1.
Author Response
Dear Reviewer,
Thank you for your comments, suggestions, and time devoted to evaluate our manuscript.
The authors deeply investigated the different kinetic release behaviour of doxorubicin and 9-PBThACl from DPPC-based liposome. Some differences were observed and discussed. Of course, this information could be of interest to select an appropriate anticancer delivery system. However, there is an important aspect that the authors need to evaluate. The design and optimization of drug delivery system should be performed also considering the particle size and zeta potential (doi.org/10.1016/j.colsurfb.2021.112217). Both of these properties influence the cellular uptake and also the stability in physiological environment. Therefore, in my opinion, the have to better characterize their system and results discussed in light of both the particles dimension and superficial charges.
Following your suggestion, better characterization of the liposomal complexes has been provided at the beginning of Results&Discussion section with the liposome size distribution, as shown in Figure 1. The details of the liposomes size measurements and stability studies were additionally reported in Materials&Methods section (subsections 3.5 and 3.6).
In the Materials and Methods section:
- The authors sinthetised the 9-PBThACl compound. Where is the structural characterization? In particular the authors have to show the NMR spectra and HPLC mass data.
The entire structural characterization of 9-PBThACl compound, including the NMR spectra and HPLC mass data, was reported in Ref. 16 [Zięba, A.; Latocha, M.; Sochanik, A.; Nycz, A.; Kuśmierz, D. Synthesis and in vitro antiproliferative activity of novel phenyl ring-substituted 5-alkyl-12(H)-quino[3,4-b][1,4]benzothiazine derivatives. Molecules 2016, 21(11), 1455.], as indicated in the Reference section of the manuscript.
- The figure of 9-PBThACl (Fig. 8) should not be present in this section since the it is already shown in the Scheme 1.
Following your suggestion Figure 8 has been deleted from the manuscript.
Reviewer 2 Report
First of all, congratulations on the good work done. It is perfectly structured and very easy to understand. I do not know if these liposomes have been characterized in any previous article, in terms of size, zeta potential, etc., in addition to the EE% and DL.
I would like to solve some questions and provide some suggestion.
Thanks in advance.
1. In the materials and methods section, section 3.3, was the size of the liposomes homogenized by any method? What size were the liposomes?
2. In section 3.6. what was the final dialysis time to remove the non-encapsulated drug? And at what predetermined times was the buffer changing?
3. Why was the temperature of 41°C selected to carry out the drug release assay?
4. Tables 1-4 do not include S.D., what was the number of samples n=?
5. In section 2.2, specifically lines 137-140, in my opinion they should be in the material and methods section. Because I have not found any section in that section that describes how the stability of the liposomes was evaluated.
6. In addition to evaluating the stability of the liposomes in terms of release (DL) and EE%, should any other parameter, appearance, growth of bacteria or fungi in the aqueous medium be determined?
7. Has any method been proposed to stabilize the liposomes? For example, freeze-drying.
8. In figure 1, I imagine that the average curves of each formulation are represented at different pHs, what is the value of n? And the same in figures 4 and 6.
Author Response
Dear Reviewer,
Thank you for your comments, suggestions, and time devoted to evaluate our manuscript.
First of all, congratulations on the good work done. It is perfectly structured and very easy to understand. I do not know if these liposomes have been characterized in any previous article, in terms of size, zeta potential, etc., in addition to the EE% and DL.
I would like to solve some questions and provide some suggestion.
Thanks in advance.
The applied methodology of liposome preparation is our authorial procedure that is based on the modified Papahadjopoulo’s method described in details in Ref. [41]. The uniqueness of the methodology relies on the quick and economical liposome’s preparation with the desired size of approximately 100 nm. The physicochemical properties of the generated liposomes have been characterized and described in the set of papers by Danuta Pentak et al. (please, see Ref. [2,3,14,27,28,47,49], etc.). The employed empirical methods, including the transmission electron microscope (TEM), the atomic force microscope (AFM), the nanoparticle tracking analysis (NTA) or the nuclear magnetic resonance (NMR) confirmed many times that the proposed methodology allows to generate small (approximately 100 nm), liposomal monolayer.
Following your suggestion, better characterization of the liposomal complexes has been provided at the beginning of Results&Discussion section with the liposome size distribution, as shown in Figure 1. The details of the liposomes size measurements and stability studies were reported in Materials&Methods section (see subsections 3.5 and 3.6).
- In the materials and methods section, section 3.3, was the size of the liposomes homogenized by any method? What size were the liposomes?
The following statement has been included into section 3.3:
“In order to unify the dimensionality of the analyzed structures the standard method of homogenization was applied using the Avanti mini extruder with 100 nm filter.” - In section 3.6. what was the final dialysis time to remove the non-encapsulated drug? And at what predetermined times was the buffer changing?
In order to separate the non-encapsulated drug molecule from the liposomal complexes the multistage dialysis method has been applied. In practice, the dialysis was run pending the lack of signal detection of the investigated compounds in the supernatant using the UV/Vis method. The presented findings were preceded by four-stage dialysis – each lasting one hour. - Why was the temperature of 41°C selected to carry out the drug release assay?
1,2-Dipalmitoyl-sn-glycero-3-phosphocholine (DPPC) is the main component of the phospholipid membrane with the point of the main phase transition at temperature of ≈ 41oC. Obviously, the temperature of the phase transition is dependent on the length of the hydrocarbon chains and type of the lipid polar groups. In case of the saturated phospholipids, the temperature of the phase transition grows with the increment of the acyl group’s length. At ≈ 41oC the main phase transition TC in DPPC layers is observed while the melting of the acyl chains appears, then La structure is generated, in which the chains are not assembled into two-dimensional net and are corrugated thanks to trans-gauche conformational shifts. The molecular surface of the single hydrocarbon chain changes gradually to 0.239 nm2. Moreover, the reduction of the layer gauge and the structural period of repeatability in the liquid crystal are observed thanks to the hydrocarbon chain corrugation, as was recorded using X ray method [M. Kriechbaum; P. Laggner, States of phase transitions in biological structures, Progress in Surface Science, 51, 233-261 (1996)]. The La structure is frequently called the liquid (crystalline) phase, while LC, Lb’ and Pb’ are named as gels, respectively. In other words, the fluidity of the La phase is higher compared to the other phases. In fact, the La phase is characterized by relatively huge lateral diffusion coefficient, small viscosity and the lack of the acyl chain corrugation. In this context, the phase transition Pb’ à La is the main phase transition due to the fluidity of the membrane, the increased cooperatives and the heat of transition compared to sub- and over-transition [Ch. Huang; S. Li, Calorimetric and molecular mechanics studies of the thermotropic phase behavior of membrane phospholipids, Biochimica et Biophysica Acta, 1422, 273-307 (1999)].
In summary, the fluidity of the membrane bilayer is the highest at temperature of ≈41oC; therefore, the increased process of the liposome-encapsulated drug release is being observed. Moreover, the elevated temperature may imitate the inflammatory conditions of living organisms, especially in the tumor surroundings. - Tables 1-4 do not include S.D., what was the number of samples n=?
Tables 1-4 have been corrected according to your recommendation. All experiments and measurements were conducted in triplicate (n=3) and the data were reported as a standard deviations (±std). - In section 2.2, specifically lines 137-140, in my opinion they should be in the material and methods section. Because I have not found any section in that section that describes how the stability of the liposomes was evaluated.
The specified lines 137-140 have been moved to Materials&Methods section as your suggested. - In addition to evaluating the stability of the liposomes in terms of release (DL) and EE%, should any other parameter, appearance, growth of bacteria or fungi in the aqueous medium be determined?
In the current paper the assessment of the liposomes stability in terms of release (DL) and EE% was conducted, because the investigated compounds revealed the anticancer activities. Regarding your suggestion, better characterization of the liposomal complexes has been provided at the beginning of Results&Discussion section with the liposome size distribution, as shown in Figure 1. The details of the liposomes size measurements and stability studies were reported in Materials&Methods section (subsections 3.5 and 3.6), but your valuable suggestion to provide wider liposomal characteristics will be definitely taken into consideration in the next parts of our reports. In fact, we have not performed the additional evaluation of any other parameters like appearance, growth of bacteria or fungi in the aqueous medium, but we continue our research as a series of experiments, e.g., using the liposomal complexes with human serum albumin (HSA) and compounds showing antibacterial (S. aureus MRSA SA 630, S. aureus MRSA SA 3032, S. aureus MRSA SA 63718, S. aureus Sa ATCC 29213, Escherichia coli) and antifungal (C. albicans CCM 8261, C. krusei CCM 8271, C. parapsilosis CCM 8260) properties, respectively. - Has any method been proposed to stabilize the liposomes? For example, freeze-drying.
At that moment, the main objective of the presented methodology is the detailed characteristics of the liposomal complexes as carriers for a new group of phenyl ring-substituted 5-alkyl-12(H)-quino[3,4-b][1,4]benzothiazine derivatives. In Authors’ mind, the whole bunch of experiments is being performed, e.g., including the interactions of liposomal complexes with human serum albumin (HSA). The Reviewer’s suggestion to stabilize the liposomes using freeze-drying method will be definitely taken into consideration in the current research. - In figure 1, I imagine that the average curves of each formulation are represented at different pHs, what is the value of n? And the same in figures 4 and 6.
The presented in Figures 1,2, and 4 findings report the changes of LDPPC/9-PBThACl, LDPPC/DOX, LDPPC/9-PBThACl/DOX liposomal complexes in a function of the environment pH variations (pH=5.5, 6.0, 6.5, 7.4). In each case the experiment was performed in triplicate (n=3). The standard deviation (±std) was presented in Figures 2,4, 5 and the numerical ±std values were also reported in Tables 1-4, respectively.
Round 2
Reviewer 1 Report
The authors made all required changes. The manuscript is now suitable for the publication.
Minor editing of English language required